# OpenReview forum: "What LLMs Explain Is Not What They Believe: Evaluating Explanation Sufficiency Under Models' Own Input Beliefs"
_ICML.cc/2026/Conference — ICML 2026 regular_

### Official Review · Reviewer_7NHV · 2026-03-12

**Soundness:** 2
**Presentation:** 3
**Significance:** 2
**Originality:** 2
**Overall Recommendation:** 3
**Confidence:** 3

**Summary:**

This paper studies how to evaluate whether free-text explanations produced by large language models are actually sufficient to justify the model’s predictions. The authors generalize the classical notion of sufficiency, originally developed for feature attribution methods, to arbitrary explanations such as natural language rationales. They show that explanation sufficiency is inherently relative to the input distribution and therefore cannot be meaningfully defined without specifying which alternative inputs are considered possible. Based on this observation, the paper proposes a notion of self-consistent sufficiency, where the input distribution is induced by the LLM itself: the model generates alternative inputs conditioned on the explanation, representing inputs that could plausibly produce the same explanation. The paper then introduces SCSuff, an information-theoretic metric based on the KL divergence between the model’s prediction distribution for the original input and the prediction distribution averaged over explanation-conditioned alternatives. Experiments on nine LLMs across four datasets show that SCSuff broadly aligns with targeted perturbation tests when those are applicable, while also providing signals in settings where existing tests are inconclusive. The results further suggest that explanation sufficiency varies with the induced input distribution and does not correlate strongly with model size, accuracy, or predictive uncertainty.

**Compliance With Llm Reviewing Policy:**

Affirmed.

**Final Justification:**

The paper is clearly written, well motivated, and addresses a timely question, with a clean framing of explanation sufficiency for free-text LLM explanations and a principled metric that is supported by reasonably broad experiments. However, I remain unconvinced that the overall contribution is strong enough for acceptance: the method mainly measures consistency within the model’s own belief space because the same model is used to generate predictions, explanations, and alternative inputs, which limits its broader significance as an explanation-quality measure; in addition, the theoretical contribution feels conceptually straightforward, and some concerns remain about the reliability of the generated alternatives and dependence on the alteration process. The rebuttal was helpful in clarifying the intended scope of the work and partially addressed robustness concerns through additional sensitivity analyses, but it did not substantially change my assessment of the paper’s originality or significance. Overall, I view the work as thoughtful and potentially useful, but still somewhat incremental and borderline for acceptance, so I keep my original score unchanged

**Key Questions For Authors:**

The method relies on the same model to generate predictions, explanations, and alternative inputs. Have you tested whether SCSuff changes significantly when the alternative inputs are generated by a different model?

How do you verify that the generated alternative inputs truly satisfy the constraints implied by the explanation? If some alternatives violate the explanation, this could affect the sufficiency estimate.

The results show strong sensitivity to the alteration prompt used to generate alternative inputs. Could you provide a more systematic analysis of how robust SCSuff is to different prompt designs?

Since the estimator relies on sampling alternative inputs and computing log probabilities, what is the observed variance of SCSuff across different sampling runs or random seeds?

How should practitioners interpret low SCSuff values in practice? Do they necessarily indicate poor explanations, or could they reflect properties of the induced input distribution rather than the explanation itself?

**Limitations:**

yes

**Strengths And Weaknesses:**

The paper tackles a timely problem, namely how to assess whether free-text explanations from LLMs actually capture the information needed to reproduce the model’s prediction. The conceptual framing is clean and well motivated: extending sufficiency from feature subsets to arbitrary explanations is a natural step, and the observation that sufficiency depends on the assumed input distribution is both intuitive and theoretically clarified in the paper. The proposed SCSuff metric is principled and easy to implement given access to model likelihoods, and the empirical evaluation is reasonably broad, covering multiple model families and datasets. The experiments illustrating how the metric changes when the induced input distribution is altered are particularly compelling and support the central claim that sufficiency is distribution-relative.

At the same time, there are several limitations that make the contribution feel somewhat borderline. First, the evaluation setup relies heavily on the same model to produce the prediction, the explanation, and the alternative inputs. This introduces a form of circularity: the metric essentially measures consistency within the model’s own belief space, which may limit how informative it is about explanation quality in a broader sense. Second, while the theoretical result about distribution-relative sufficiency is correct, it is conceptually straightforward and does not add substantial technical depth beyond formalizing the intuition. Third, the experimental setup leaves a few open questions. In particular, it is not clear how reliably the generated alternative inputs preserve the constraints described in the explanation, and the results could be sensitive to the prompt used to generate these alternatives. Finally, the metric primarily evaluates output-level consistency rather than faithfulness of internal reasoning, so it should probably be interpreted as a diagnostic for model self-consistency rather than a general explanation faithfulness measure.

Overall, the work provides an interesting perspective on explanation sufficiency and introduces a potentially useful diagnostic tool, but the reliance on model-induced input distributions and the limited depth of the theoretical results make the overall contribution somewhat incremental.

---

> ### Author Rebuttal · Authors · 2026-03-31
>
> We thank the reviewer for their feedback and positive assessment of our framing, metric, and experiments. We address your concerns below to clarify the scope and robustness of our work.
> ## Using the same model for predictions, explanations, and alternative inputs
> The reviewer notes that using the same model may limit SCSuff's assessment of broader explanation quality. This is by design, as SCSuff measures **self-consistency**. Self-consistency is a necessary condition for an LLM to explain its behavior truthfully [1], making SCSuff one, but not only, important assessment of explanation quality. Prior work [2] also suggests LLMs have privileged access to their internal mechanisms, so we focus on measuring how self-consistently sufficient LLM explanations are given this access. Future work could study sufficiency under human-induced input distributions or inputs generated by other LLMs.
>
> We additionally measure SCSuff on Qwen and Llama using alternatives generated by Ministral. SCSuff varies across generators (max difference 0.175), and in 3/4 cases, using the same model yields higher SCSuff, indicating explanations are more consistent with alternatives from the same model. This supports the reason for our focus on self-consistency.
>
> Table 1: SCSuff with alternatives from different LLM
> |Model|Dataset|Same model|Different model|
> |-|-|-|-|
> |Qwen3-8B|MMLU + authority|0.354|0.179|
> ||IMDB|0.614|0.594|
> |Llama3.1-8B|MMLU + authority|0.232|0.290|
> ||IMDB|0.810|0.805|
> ## Alternatives satisfy explanation constraints
> Verifying whether generated alternatives satisfy explanation constraints would necessarily **impose external (e.g., human) assumptions** on the input distribution. Our metric measures self-consistency, and what the model considers consistent may differ from human expectations. Thus, such verification is not included by design.
>
> To ensure the LLMs do not ignore explanation constraints when generating alternatives, we do include qualitative examples in the appendix. We additionally use GPT-5-mini as an evaluator to assign binary scores for whether alternatives preserve explanation constraints for 150 samples. 70% of generated alternatives are judged to be consistent, suggesting that LLMs did use the explanation constraints to generate alternative inputs.
> ## Sensitivity analyses
> **Alteration prompt:** Deliberately including contradictory few-shots in the alteration prompt can change SCSuff substantially (Section 6.2), but typical paraphrasing of the prompt does not (max difference 0.028), indicating that SCSuff is not sensitive to exact wording of alteration prompt.
>
> **Random seed:** Across 2 additional seeds, SCSuff remains stable (max difference 0.046), demonstrating that SCSuff scores are reproducible across sampling runs.
>
> Table 2: Sensitivity of SCSuff
> |Model|Dataset|Original|Paraphrase|Seed 2|Seed 3|
> |-|-|-|-|-|-|
> |Qwen3-8B|MMLU + authority|0.354|0.326|0.328|0.308|
> ||IMDB|0.614|0.591|0.591|0.597|
> |Llama3.1-8B|MMLU + authority|0.232|0.222|0.257|0.227|
> ||IMDB|0.810|0.812|0.823|0.796|
> ## Practical interpretation of SCSuff
> A low SCSuff indicates either that the explanation omits important information the LLM uses for its prediction, or that the generated alternatives do not align with the model’s internal beliefs. Both reflect a lack of self-consistent sufficiency, either in the explanation or the induced input distribution. Self-consistent sufficiency is a necessary condition for an LLM to explain its behavior sufficiently and truthfully, so a low SCSuff indicates a poor ability of the LLM to explain its own behavior, regardless of the source of inconsistency.
> ## Theoretical contribution
> The reviewer notes that conceptualizing distribution-relative sufficiency is straightforward. Explicit formalization is nevertheless necessary for LLM explanations where input distributions are not specified: the formalization, simple as it may be, makes clear why defining the input distribution is essential. In contrast, for feature attributions, input distributions are given by models’ training distributions. Our formulation provides a principled way to generalize sufficiency to arbitrary explanations.
> ## Consistency vs Faithfulness
> It is true that output-level consistency differs from internal reasoning faithfulness. Consistency verifies alignment between text explanations and model behavior in black-box settings, but it does not assess internal reasoning. Faithfulness assesses the model’s internal mechanism, but interpreting complex model internals in human-understandable terms remains challenging. While both are desirable, we follow prior work [1] in focusing on output-level consistency in a black-box setting. Thus, we explicitly frame SCSuff as a consistency metric and avoid claims about internal faithfulness.
> ## References
> [1] https://aclanthology.org/2024.acl-long.329/
>
> [2] https://arxiv.org/abs/2511.08579
> ***
> **If our responses resolve your concerns, we would appreciate your consideration in raising your score.**

---

> > ### Author Rebuttal · Reviewer_7NHV · 2026-04-04
> >
> > Thank you for the rebuttal. The added sensitivity analysis is helpful, especially the results on prompt paraphrases and additional random seeds, which partially address my concern about robustness. I also appreciate the clarification that the intended scope is self-consistent sufficiency rather than a broader notion of explanation faithfulness.
> >
> > That said, my main concerns remain only partially resolved. The response on the circularity issue mostly clarifies the intended scope, but does not change my view that the metric mainly captures consistency within the model’s own belief space. The cross-generator comparison is useful, but still limited in scale. Likewise, the added check on whether alternatives satisfy explanation constraints is a helpful sanity check, but it does not fully remove my concern about the reliability of the alteration process.
> >
> > Overall, the rebuttal improves the clarity of the paper, but it does not substantially change my assessment that the contribution is interesting yet somewhat incremental and still borderline for acceptance. I therefore keep my score unchanged.

---

> > > ### Author Response · Authors · 2026-04-06
> > >
> > > We thank the reviewer for their constructive engagement with our rebuttal.
> > >
> > > Regarding the concern about circularity, we respectfully argue that "circularity" is not the right characterization here. Circularity implies that an argument is not valid - that the conclusion is smuggled into the premise. That is not the case in our setup. Using the same model to generate both the explanation and the alternative inputs is not a logical flaw; rather, as you acknowledged, it is precisely the correct way to measure self-consistency, which is the explicit goal of this paper. There is no hidden assumption being validated by itself.
> > >
> > > We also want to emphasize that the framework is not inherently restricted to self-consistency - it can readily accommodate alternatives generated by a different model, as our cross-generator experiments demonstrate. We view those results as preliminary evidence of this generality, and we agree that a more thorough investigation would be a valuable direction for future work.
> > >
> > > If our response addresses your concerns, we would appreciate your consideration in raising your score. Please let us know if any further clarification would help

---

### Official Review · Reviewer_iu1F · 2026-03-12

**Soundness:** 3
**Presentation:** 3
**Significance:** 3
**Originality:** 3
**Overall Recommendation:** 4
**Confidence:** 4

**Summary:**

This paper presents a novel perspective on evaluating the sufficiency of free-text explanations generated by LLMs. The authors argue that most existing methods primarily measure comprehensiveness and are limited by their lack of generalizability. Instead of perturbing the inputs, the authors propose a new evaluation paradigm that leverages the explanation itself to generate alternative inputs within the same distribution. The underlying premise is that a sufficient explanation should ensure consistent model outputs across all alternative inputs within this distribution. Extensive experiments broadly validate the authors' hypotheses.

**Compliance With Llm Reviewing Policy:**

Affirmed.

**Final Justification:**

I appreciate the authors' response to all my questions. The rebuttal has reinforced my prior assessment and I will keep my positive evaluation.

**Key Questions For Authors:**

My understanding of "comprehensiveness" is that it evaluates whether the explanation includes all input features predictive of the output. Based on this, could we assume that an explanation satisfying comprehensiveness will inevitably satisfy sufficiency? However, this assumption appears to conflict with some of your experimental conclusions. Could you clarify the theoretical boundary between these two concepts in your framework?

**Limitations:**

yes

**Strengths And Weaknesses:**

Strengths
1. The paper introduces an intriguing and novel definition of sufficiency, representing a significant step forward compared to prior research.
2. The proposed methodology is concise, easy to implement, and highly generalizable across different models and datasets.

Weaknesses
1. As noted by the authors, the more constraints an explanation imposes on the input, the more restricted and sparse the corresponding input distribution becomes. I believe the trade-off between the "degree of input restriction imposed by the explanation" and the "degrees of freedom of the input distribution" is a critical factor influencing the proposed metric. For example, if an explanation already satisfies sufficiency but additionally restricts extraneous input features, the distribution space of alternative inputs will shrink further. Does this imply that the proposed metric lacks the capability to identify the "minimal sufficient set" (or the most concise sufficient explanation)?
2. Building on the first point, the properties of this metric would be elucidated more clearly if the experimental section included comparative results from intervening in or masking parts of the explanations.

---

> ### Author Rebuttal · Authors · 2026-03-31
>
> We thank the reviewer for highlighting the novelty of our sufficiency definition and the generality of our method. We address your questions below.
> ## Sufficient vs. Minimally sufficient
> The reviewer asks if SCSuff can detect when an explanation is redundant, e.g., restricts extraneous input features, effectively fixing the entire input. As noted in our limitations, such explanations are **trivially sufficient**, thus represent a degenerate case. This degenerate case is shared by prior sufficiency metrics in feature attributions; thus, these metrics are often paired with measures of explanation complexity [1]. While this paper focuses on defining and evaluating explanation sufficiency alone, future work could jointly optimize SCSuff and a complexity metric to produce minimally sufficient free-text explanations.
>
> To test whether this concern is practically relevant, we measure the proportion of the original input retained in the alternatives across 3 models and 2 datasets using exact matching (if the explanation is trivial, the alternatives are expected to be very similar). Table 1 shows overlap is moderate, indicating alternatives generated in practice are meaningfully perturbed and not degenerate copies of the original input.
>
> Table 1: Distribution of proportion of the original input retained in the alternatives
> |min|25th|50th|75th|max|mean|std|
> |-|-|-|-|-|-|-|
> |0.002|0.392|0.452|0.517|1.000|0.441|0.146|
> ## Comprehensiveness vs. Sufficiency
> The reviewer correctly notes that a fully comprehensive explanation - i.e., one that includes all input features informative of the output - is necessarily sufficient - i.e., contains enough information to predict the output. In our experiments (Section 6.1), self-counterfactual explanation (SCE) tests for comprehensiveness and SCSuff tests for sufficiency. However, they are computed on different explanations, as SCE evaluates counterfactual explanations (inputs that would change the output), whereas SCSuff is applied to arbitrary free-text explanations (chain of thought in our experiments). Therefore, we do not expect a direct correspondence between their scores. The difference between SCE and SCSuff dataset-level scores only indicates that an LLM’s ability to generate good counterfactual inputs does not imply its ability to generate sufficient free-text explanations and vice versa.
>
> Causal Concept Faithfulness (CCF) is another method that tests for comprehensiveness. We did not perform experiments with CCF due to its heavy reliance on powerful LLMs, and thus, evaluating CCF is costly. If CCF and SCSuff were computed on the same explanations, we expect explanations with high (close to optimal) CCF to also have high SCSuff, since full comprehensiveness implies sufficiency.
> ## References
> [1] https://pubmed.ncbi.nlm.nih.gov/33954293/
> ***
> **If our responses resolve your concerns, we would appreciate your consideration in raising your score.**

---

> > ### Author Rebuttal · Reviewer_iu1F · 2026-04-01
> >
> > I appreciate the authors' response to my questions. The authors mention that they expect high CCF scores to correlate with high SCSuff when computed on the same explanations. Out of curiosity, is it feasible to perform even a small-scale verification of this expectation? But regardless of having this result or not, I will maintain my positive evaluation.

---

> > > ### Author Response · Authors · 2026-04-06
> > >
> > > We appreciate your continued positive evaluation and agree that verifying the relationship between comprehensiveness and sufficiency metrics would be interesting.
> > >
> > > To conduct a small-scale experiment, we focus on GPT-3.5, as it is the only model evaluated in the CCF paper [1] that provides log probabilities. We randomly selected 10 out of 30 total BBQ input samples used in the CCF paper, and sampled 5 chain-of-thoughts (CoTs) per input sample. We use the same CoTs for both CCF and SCSuff, ensuring a direct comparison over the same 50 input-explanation pairs.
> > >
> > > Using the public CCF code repository [2], we recompute CCF on these 50 pairs of input and CoT. For SCSuff, we adapt our method to GPT-3.5 and generate 5 alternatives per input-CoT pair. Results are shown below.
> > >
> > > |Sample ID|CCF|SCSuff|
> > > |-|-|-|
> > > |183|0.786|0.903|
> > > |578|0.298|0.763|
> > > |726|0.783|0.968|
> > > |913|0.993|0.906|
> > > |1058|0.820|0.757|
> > > |1675|0.257|0.848|
> > > |1714|0.663|0.177|
> > > |1844|1.000|0.988|
> > > |2056|0.470|0.873|
> > > |2209|0.888|0.667|
> > >
> > > We find that samples with near-optimal CCF (e.g., 913, 1844) also achieve high SCSuff ($\geq 0.9$), and higher CCF scores ($\geq 0.6$) generally correspond to higher SCSuff ($\geq 0.6$). This aligns with our expectation: **fully comprehensive explanations are sufficient**. At lower CCF, the relationship weakens, as less comprehensive explanations can still be sufficient. For example, if an input contains multiple redundant cues informative of the output, then an explanation that mentions only one cue can be fully sufficient but have low comprehensiveness. Overall, these results align with our expectation about the relationship between comprehensiveness and sufficiency.
> > >
> > > **If our responses address your questions, we would greatly appreciate your consideration in raising your score for our paper.**
> > >
> > > _References_
> > >
> > > [1] https://openreview.net/forum?id=4ub9gpx9xw
> > >
> > > [2] https://github.com/kmatton/walk-the-talk

---

### Official Review · Reviewer_UZTZ · 2026-03-13

**Soundness:** 3
**Presentation:** 4
**Significance:** 3
**Originality:** 3
**Overall Recommendation:** 4
**Confidence:** 4

**Summary:**

The paper works on the challenge of evaluating the "sufficiency" of free-text explanations generated by Large Language Models (LLMs). It theoretically generalizes the classical notion of feature attribution sufficiency to arbitrary explanations, and formally proves that explanation sufficiency is inherently relative to the underlying input distribution. To apply this for LLMs, the authors propose using the LLM itself to generate alternative inputs conditioned on a given explanation, and capturing the model's own induced input distribution. Based on this, they introduce SCSuff, an information-theoretic metric to quantify self-consistent sufficiency. Experiments across multiple LLMs and datasets demonstrate that SCSuff aligns with targeted perturbation tests, that sufficiency is indeed relative to the input distribution, and that current LLM explanations are generally insufficient regardless of model scale or accuracy.

**Compliance With Llm Reviewing Policy:**

Affirmed.

**Final Justification:**

Thanks to the authors for the responses. I will maintain my positive evaluation of the paper.

**Key Questions For Authors:**

1. How does SCSuff behave empirically when an LLM is prompted to give extremely verbose explanations that inadvertently copy most of the input? Does the generation of alternative inputs collapse, and if so, how can the metric distinguish between "highly sufficient" and "highly redundant" explanations?

2. How do you verify that the generated alternative inputs truly preserve the constraints of the explanation?

3. In your log-probability computation, the final answer relies on the specific decoding path of the generated CoT. Did you try computing the answer distribution probabilities directly (answer-only likelihood, without the CoT in the context window) to assess how much the CoT itself artificially influences or brittles the SCSuff score?

**Limitations:**

yes

**Strengths And Weaknesses:**

Strengths:

1. The theoretical formalization (Theorem 4.3) is sound and elegantly extends classical feature attribution sufficiency to free-text rationales. The proposed metric, SCSuff, is well-grounded in information theory and avoids the heavy reliance on predefined biases or shortcuts that plague many existing perturbation-based metrics.

2. The paper is well-structured and written. The motivation is clear, and Figure 1 effectively builds intuition for how alternative inputs can remain compatible with an explanation while altering the output. The mathematical derivations in Section 4 are rigorous and easy to follow.

3. Moving beyond task-specific perturbation tests to a generalized, model-agnostic metric is a significant step forward for the interpretability and alignment communities. The finding that scaling up model size does not organically improve self-consistent sufficiency is an important empirical result for the field.

Weaknesses:

1. From what I understood, there is a circularity in the evaluation loop. Using the exact same model to predict the answer, write the explanation, and generate the alternative inputs risks reward-hacking, where the model simply produces an echo chamber of its own biases rather than a truly sufficient explanation. Furthermore, the normalization equation used to bound the SCSuff score between 0 and 1 is heuristic. Its behavior could become highly erratic or artificially inflated when the model's output probability distribution is nearly deterministic.

2. While the metric is conceptually strong, the lack of verification regarding whether the generated alternative inputs actually preserve the explanation's constraints slightly dilutes the confidence in the final dataset-level scores.

3. The methodology for computing log-probabilities relies on measuring the probability of the Final Answer after the model has generated a deterministic CoT. Because LLMs are autoregressive, this conflates the true probability of the answer with the specific decoding path of the CoT, which could introduce a confounding variable into the SCSuff scores.

---

> ### Author Rebuttal · Authors · 2026-03-31
>
> We thank the reviewer for their feedback and positive assessment. We address your concerns below to clarify the scope and implementation of our method.
> ## Redundant explanations
> The reviewer asks how SCSuff behaves for verbose explanations that fix the input. As noted in limitations, such explanations are **trivially sufficient**, representing a degenerate case. This is shared by prior sufficiency metrics, so they are often paired with measures of complexity [1]. While this paper focuses on defining and evaluating explanation sufficiency alone, future work could jointly optimize SCSuff and complexity metric to produce minimally sufficient free-text explanations.
>
> To test this concern in practice, we measure the proportion of the input retained in the alternatives across 3 models and 2 datasets using exact matching, as trivial explanations yield similar alternatives. Table 1 shows overlap is moderate, indicating alternatives generated in practice are meaningfully perturbed and not degenerate copies of the original input.
>
> Table 1: Proportion of input preserved in alternatives
> |min|25th|50th|75th|max|mean|std|
> |-|-|-|-|-|-|-|
> |0.002|0.392|0.452|0.517|1.000|0.441|0.146|
> ## Alternatives satisfy explanation constraints
> Verifying whether generated alternatives satisfy explanation constraints would **impose external assumptions** on the input distribution. Our metric measures self-consistency, and what the model considers consistent may differ from human expectations. Thus, such verification is not included by design.
>
> To ensure the LLMs do not ignore explanation constraints when generating alternatives, we include qualitative examples in the appendix. We additionally use GPT-5-mini to assign binary scores for whether alternatives preserve explanation constraints for 150 samples. 70% of generated alternatives are judged to be consistent, suggesting that LLMs use the explanation constraints to generate alternatives.
> ## Output probability with CoT
> The reviewer asks if conditioning on CoT affects output probability estimates. We compute log probabilities with CoT by design, as part of our choice of predictive model. Our sufficiency definition requires 3 components: predictive model $q(y|x)$, explanation $e$, and alternative input distributions $q(x'|e)$, with no restriction on how each is obtained. In particular, $q(y|x)$ can be an LLM’s prediction with or without CoT, corresponding to **different predictive distributions**. We focus on CoT-based predictions and use CoT as the explanation in the main text, as this reflects typical LLM usage. Evaluating sufficiency with $q(y|x)$ without CoT is a different possible choice, where explanations can be obtained post hoc.
>
> We perform additional experiments using $q(y|x)$ without CoT and observe modest differences (max difference 0.074), suggesting that these LLMs can explain their direct answers without CoT at a similar self-consistent sufficiency level.
>
> Table 2: SCSuff with different $q(y|x)$
> |Model|Dataset|With CoT|Without CoT|
> |-|-|-|-|
> |Qwen3-8B|MMLU + authority|0.354|0.339|
> ||IMDB|0.614|0.540|
> |Llama3.1-8B|MMLU + authority|0.232|0.237|
> ||IMDB|0.810|0.825|
> ## Using the same model for predictions, explanations, and alternative inputs
> The reviewer raises a concern about using the same model, similar to Reviewer 7NHV. This is by design, as SCSuff measures self-consistency [2]. Due to space, we refer the reviewer to our response to Reviewer 7NHV for details, including additional results with alternatives generated by another LLM. We acknowledge the potential for reward-hacking, e.g., explanations encoding information only the model can decode. While this may differ from human expectations, it does not affect self-consistency. Future work could further study sufficiency under human-induced input distribution or alternative inputs generated by another LLM.
> ## Normalization
> The reviewer raises a concern about the normalization used to bound SCSuff. This follows common practice in normalized signal comparisons [3]. However, when the negative log likelihoods (NLL) $\mathcal{L}(y|x)$ and $\mathcal{L}(y|x’)$ are near 0, the explanation should be sufficient, but the normalization term can be small and artificially lower SCSuff. In practice, however, SCSuff remains high: for samples with both NLL < 0.01 across 3 models and 2 datasets, the median SCSuff is 0.895 and interquantile range (0.604, 1.0), significantly exceeding the average over all samples 0.381. This shows that normalization may lower SCSuff in some cases but does not significantly affect overall interpretation of our metric. Future work could further stabilize the metric, e.g., adding an epsilon numerator and denominator or normalizing by output entropy.
> ## References
> [1] https://pubmed.ncbi.nlm.nih.gov/33954293/
>
> [2] https://aclanthology.org/2024.acl-long.329/
>
> [3] https://ntrs.nasa.gov/citations/19740022614
> ***
> **If our responses resolve your concerns, we would appreciate your consideration in raising your score.**

---

> > ### Author Rebuttal · Reviewer_UZTZ · 2026-04-04
> >
> > The authors have answered the questions I had. Originality is OK. I will maintain my positive score.

---

### Decision · Program_Chairs · 2026-04-30

**Decision:**

Accept (regular)

**Comment:**

This paper proposes a new perspective to evaluate the sufficiency of free-text explanations generated by LLMs by leveraging the explanation itself to generate alternative inputs within the same distribution. Most reviewers found that this paper is well-written with clear motivation, and the proposed method is easy to implement and generalizable across models and dataset. While there are some concerns regarding the evaluation and circularity, it seems to be addressed by the authors in the rebuttal.